# Greedy Relaxations of the Sparsest Permutation Algorithm

**Wai-Yin Lam**[1]        **Bryan Andrews**[1]        **Joseph Ramsey**[1]

[1] Department of Philosophy, Carnegie Mellon University, Pittsburgh, Pennsylvania, USA

## Abstract

There has been an increasing interest in methods that exploit permutation reasoning to search for directed acyclic causal models, including the "Ordering Search" of Teyssier and Kohler and GSP of Solus, Wang and Uhler. We extend the methods of the latter by a permutation-based operation *tuck*, and develop a class of algorithms, namely GRaSP, that are computationally efficient and pointwise consistent under increasingly weaker assumptions than faithfulness. The most relaxed form of GRaSP outperforms many state-of-the-art causal search algorithms in simulation, allowing efficient and accurate search even for dense graphs and graphs with more than 100 variables.

## 1 INTRODUCTION

Searching for causal models by identifying patterns of conditional independence in observational data has become a well-established activity, though it is not without detractors. For one thing, it is commonly believed that the only correct method for establishing causal relationships is through experimental manipulation, as is done in a randomized controlled trial. Accordingly, causal inference from observational data alone can be seen as second-rate. This is not completely unreasonable; many causal search algorithms, even in seemingly ideal conditions at reasonable sample sizes, demonstrate poor performance, calling into question whether their inferences can be relied upon. Furthermore, the theoretical assumptions made by these algorithms are often criticized for being too strong. More specifically, these algorithms assume that the true model belongs to a model class with no latent variables or no cycles, and that the patterns of conditional independence in the data generating distribution can be represented by the assumed model class exactly. The latter assumption is called the causal faithful-

ness condition, or faithfulness in short, and can be violated (or almost violated) by unexpected patterns of conditional independence that arise from subtleties in the distribution, such as (near) determinism or (almost) path cancellation.

The most common model class assumed by causal search algorithms is characterized by directed acyclic graphs (DAGs). Many algorithms for causal inference search the space of causal DAGs, such as the PC ("Peter and Clark", [Spirtes et al., 2000]) and GES ("Greedy equivalence Search", [Chickering, 2002]) algorithms, and provably return a set of DAGs that contains the true model under faithfulness. However, as depicted in Section 5.2, the performance statistics of these algorithms are extremely slow to converge, especially when the true model is densely connected. One hypothesis for this phenomenon is that almost-violations of faithfulness frequently occur and impede search procedures [Uhler et al., 2013]. Accordingly, the performance of these algorithms might be improved by relaxing faithfulness, as is done by the SP ("Sparsest Permutation") algorithm of Raskutti and Uhler [2018]. SP considers the space of variable orderings and builds a DAG using a procedure inspired by Verma and Pearl [1988], where the parents of each variable are selected from the preceding variables in the permutation. Ultimately, the permutations that induce DAGs with the minimal edge count are selected.

Raskutti and Uhler [2018] proved that if the data generating distribution is a graphoid, then the set of DAGs returned by SP contains the true model asymptotically under an assumption strictly weaker than faithfulness. While SP recovers the set of all frugal models, it is super exponential in the number of variables, that is, if there are $n$ variables, then there are $n!$ permutations that must be visited. In practice, it is limited to a maximum of about nine variables due to its computational complexity. This naturally raises the question: is there an algorithm that is equally accurate in most cases for such data, but that can scale to larger problems?

Teyssier and Koller [2005] give a clever search and score procedure, "Ordering Search", over variable permutations,

*Accepted for the 38th Conference on Uncertainty in Artificial Intelligence* (UAI 2022).

pointing out that when two adjacent variables in a permutation are swapped, only local scores for the swapped variables need to be recalculated, the rest of the score calculation remains unchanged—this swapping operation is called an adjacency transposition (AT). The Ordering Search algorithm greedily traverses the space of permutations with adjacency transpositions using a hill-climbing approach, random restarts, and a tabu list. However, they do not give any consistency guarantees.

The ESP ("Edge Sparsest Permutation") algorithm of Solus et al. [2021] iterates upon the Ordering Search algorithm by greedily traversing the space of permutations by sequences of ATs where each AT leads to an equal or smaller edge count, found by depth first search (DFS), to achieve asymptotic correctness. Also, their TSP ("Triangle Sparsest Permutation") algorithm uses the theory of Chickering [2002] to navigate the space of DAGs, more efficiently than ESP, under a stronger assumption. A simulation study using a Python implementation of TSP [Solus et al., 2021] suggests that this procedure is fast, but has difficulty scaling accurately to moderate or large sized graphs [Lu et al., 2021].

To address the scaling problem for both accuracy and timing, in this paper we explore different ways of traversing the space of permutations that get closer to the performance and assumption relaxation of Raskutti and Uhler while maintaining scalability. As part of this effort, we also use the "Grow-Shrink" algorithm from Margaritis and Thrun [1999] to learn the DAG.

In what follows, we give an elaboration of the theoretical background of our set of permutation-based procedures, GRaSP ("Greedy Relaxations of Sparsest Permutation"). GRaSP has three tiers, $GRaSP_0$ (basically equivalent to TSP), $GRaSP_1$ (basically equivalent to ESP), and $GRaSP_2$ (a novel relaxation); we show how moving from a lower tier to a higher tier results in a gradual theoretical relaxation of the permutation search space and thus an improvement in accuracy. We then follow this with a study of oracle behavior for $GRaSP_0$, $GRaSP_1$, and $GRaSP_2$ on exhaustive lists of independence models with violations of faithfulness for all 4-variable regular Gaussian and positive discrete distributions and all 5-variable unfaithful DAGs with added marginal independencies between a pair of variables. We also give a detailed simulation study for the linear, Gaussian case for larger possibly dense models of up to 100 variables, with consistently accurate results using $GRaSP_2$. Further, we study an empirical example to test $GRaSP_2$. We then give a conclusion and discussion where we point out areas of immediate future work.

## 2 CONTRIBUTIONS

The most salient contribution is that $GRaSP_2$ can scale to at least 100 variables with average degree at least 10 on a laptop with high adjacency and arrowhead precision and recall for the linear, Gaussian case, addressing the long-standing practical problem of dense graph causal search in a meaningful way.

Second, theoretical assumptions required for causal discovery from previous works have been simplified, in places corrected, and reworked as a structured study of *causal razors* in Appendix D. Accordingly, the proof that $GRaSP_0$, TSP, and by implication GSP, require faithfulness is a logical discovery. Also, the proof that faithfulness is equivalent to unique Pearl-minimality is a novel contribution.

Third, we extended the discussion of unit tests initiated in [Solus et al., 2021] considerably, using the criterion that a wide variety of unit tests should systematically pass on all initial permutations using a d-separation oracle. More specifically, we run GRaSP on models detailed in Šimecek [2006a,b] and those listed in Appendix G.

Finally, the tuck operation is a novel transformation that has not been considered in the literature before. We show that traversals in the DAG-associahedron (defined in Appendix C) can be equivalently done via a tuck. Reframing TSP in terms of the tuck operation allows TSP and ESP to be neatly placed into a hierarchy. Moreover, it admits the natural generalization to $GRaSP_2$ (by not restricting which edges can be tucked).

## 3 BACKGROUND

Throughout this paper, italicized letters are used to denote variables (e.g., $X_1, Y$) and boldfaced letters for sets of variables (e.g., $\mathbf{X}$). Graphical definitions and notations related to directed acyclic graphs (DAGs) are provided in Appendix A.1. A DAG $\mathcal{G}$ over a set of measured variables $\mathbf{V} = \{X_1, ..., X_m\}$ consists of $m$ vertices $\mathbf{v} = \{1, ..., m\}$ where each vertex $i$ associates to the variable $X_i$, and each directed edge between two distinct vertices $j \rightarrow k$ represents the direct causal influence from $X_j$ to $X_k$. We write $\mathbf{i} \perp_{\mathcal{G}} \mathbf{j} \,|\, \mathbf{k}$ to denote the *d-separation* relation between $\mathbf{i}$ and $\mathbf{j}$ given $\mathbf{k}$ in $\mathcal{G}$ for any pairwise disjoint subsets of vertices $\mathbf{i}, \mathbf{j}, \mathbf{k} \subseteq \mathbf{v}$. Similarly, given a joint probability distribution $\mathcal{P}$ over $\mathbf{V}$, denote $\mathbf{X} \perp\!\!\!\perp_{\mathcal{P}} \mathbf{Y} \,|\, \mathbf{Z}$ as the *conditional independence* (CI) relation between $\mathbf{X}$ and $\mathbf{Y}$ given $\mathbf{Z}$ for any pairwise disjoint subsets of variables $\mathbf{X}, \mathbf{Y}, \mathbf{Z} \subseteq \mathbf{V}$.

A *model* is a pair $(\mathcal{G}, \mathcal{P})$ where $\mathcal{G}$ is a DAG and $\mathcal{P}$ is a joint probability distribution over the same set of measured variables $\mathbf{V}$. We use $\mathcal{G}^*$ to refer to the *true* data-generating DAG such that $(\mathcal{G}^*, \mathcal{P})$ is the true model assumed to always exist. Certain standard properties of a model can be defined in terms of the d-separation relations in $\mathcal{G}$ and the CI relations in $\mathcal{P}$. Denote $\mathtt{I}(\mathcal{G}) = \{\langle \mathbf{X_j}, \mathbf{X_k} \,|\, \mathbf{X_l}\rangle : \mathbf{j} \perp_{\mathcal{G}} \mathbf{k} \,|\, \mathbf{l}\}$ where $\mathbf{X_i} = \{X_j \in \mathbf{V} : j \in \mathbf{i}\}$ for every $\mathbf{i} \subseteq \mathbf{v}$, and $\mathtt{I}(\mathcal{P}) = \{\langle \mathbf{X}, \mathbf{Y} \,|\, \mathbf{Z}\rangle : \mathbf{X} \perp\!\!\!\perp_{\mathcal{P}} \mathbf{Y} \,|\, \mathbf{Z}\}$. Let $\mathtt{DAG}(\mathbf{V})$ be the set of all possible DAGs over $\mathbf{V}$.

**Definition 3.1** *(Markov) For any joint probability distribution $\mathcal{P}$ over $\mathbf{V}$, define $\mathtt{CMC}(\mathcal{P}) = \{\mathcal{G} \in \mathtt{DAG}(\mathbf{V}) : \mathtt{I}(\mathcal{G}) \subseteq \mathtt{I}(\mathcal{P})\}$ as the set of Markovian DAGs. $(\mathcal{G}^*, \mathcal{P})$ satisfies the Markov assumption if $\mathcal{G}^* \in \mathtt{CMC}(\mathcal{P})$.*

**Definition 3.2** *(Faithfulness) For any joint probability distribution $\mathcal{P}$, define $\mathtt{CFC}(\mathcal{P}) = \{\mathcal{G} \in \mathtt{CMC}(\mathcal{P}) : \mathtt{I}(\mathcal{P}) \subseteq \mathtt{I}(\mathcal{G})\}$ as the set of faithful DAGs. $(\mathcal{G}^*, \mathcal{P})$ satisfies the faithfulness assumption if $\mathcal{G}^* \in \mathtt{CFC}(\mathcal{P})$.*

A causal search algorithm is a procedure of recovering the causal information of the true DAG from its underlying joint probability distribution. Let $\mathtt{MEC}(\mathcal{G})$ be the *Markov equivalence class* (MEC) of $\mathcal{G}$ such that $\mathtt{I}(\mathcal{G}) = \mathtt{I}(\mathcal{G}')$ for each $\mathcal{G}' \in \mathtt{MEC}(\mathcal{G})$. One crucial goal of causal search is the identification of $\mathtt{MEC}(\mathcal{G}^*)$ from $\mathcal{P}$. With regard to this goal, a causal search algorithm is *correct* if its output DAG (or the DAG induced by its output) is in $\mathtt{MEC}(\mathcal{G}^*)$. All known causal search algorithms assume the Markov assumption, and some well-known algorithms in the relevant literature (e.g., GES) assume faithfulness as well. Nevertheless, as pointed out by Uhler et al. [2013], learning CI relations from data by hypothesis testing is error-prone, and almost-violations of faithfulness are common. This motivates the exploration of causal search algorithms which rely on assumptions strictly weaker than faithfulness. These assumptions, faithfulness included, are what we refer to as *causal razors*.

One recent approach proposed by Raskutti and Uhler [2018] is the *SP* algorithm, which identifies the set of *sparsest permutations* defined over $\mathbf{v}$ under the following causal razor. Let $\mathtt{E}(\mathcal{G})$ be the set of directed edges in a DAG $\mathcal{G}$.

**Definition 3.3** *(U-frugality) For any joint probability distribution $\mathcal{P}$, define $\mathtt{Fr}(\mathcal{P}) = \{\mathcal{G} \in \mathtt{CMC}(\mathcal{P}) : \neg\exists\mathcal{G}' \in \mathtt{CMC}(\mathcal{P})$ s.t. $|\mathtt{E}(\mathcal{G}')| < |\mathtt{E}(\mathcal{G})|\}$ and $\mathtt{uFr}(\mathcal{P}) = \{\mathcal{G} \in \mathtt{Fr}(\mathcal{P}) : \neg\exists\mathcal{G}' \in \mathtt{Fr}(\mathcal{P})$ s.t. $\mathcal{G}' \notin \mathtt{MEC}(\mathcal{G})\}$ as the sets of frugal DAGs and uniquely frugal, or u-frugal, DAGs respectively. $(\mathcal{G}^*, \mathcal{P})$ satisfies the u-frugality assumption if $\mathcal{G}^* \in \mathtt{uFr}(\mathcal{P})$.*[1]

In words, u-frugality requires that $\mathcal{G}^*$ is not only the sparsest Markovian DAG, but also that all sparsest Markovian DAGs belong to the same MEC as $\mathcal{G}^*$. Raskutti and Uhler [2018] showed that SP is correct under u-frugality which is strictly weaker than faithfulness. Below we introduce some necessary notations of permutation-based algorithms. To begin with, we refer the readers to Appendix A.2 for the *graphoid axioms*. Generally speaking, every joint probability distribution is a *semigraphoid*, strictly positive distributions are *graphoids*, and regular Gaussian distributions are *compositional graphoids*.

Given $\mathbf{V} = \{X_1, ..., X_m\}$, let $\Pi(\mathbf{v})$ be the set of all *permutations* over $\mathbf{v} = \{1, ..., m\}$. For each $\pi \in \Pi(\mathbf{v})$, let $\pi_i$

be the $i$-th vertex in $\pi$, $\pi[j]$ be the index of vertex $j$ in $\pi$ (s.t. $\pi_{\pi[j]} = j$), and $\mathtt{Pre}(j, \pi) = \{\pi_i : 1 \le i < \pi[j]\}$ be the set of vertices that precede $j$'s index in $\pi$. We say that $\pi \in \Pi(\mathbf{v})$ is a *causal order* of $\mathcal{G} \in \mathtt{DAG}(\mathbf{V})$ if $i \in \mathtt{Pre}(j, \pi)$ for each $j \in \mathbf{v}$ and each $i \in \mathtt{An}(j, \mathcal{G})$ (i.e., the set of $j$'s *ancestors* in $\mathcal{G}$). Given a graphoid $\mathcal{P}$ over $\mathbf{V}$, each $\pi \in \Pi(\mathbf{v})$ induces a DAG $\mathcal{G}_\pi$ satisfying the following condition:

$$j \in \mathtt{Pre}(k, \pi) \text{ and } X_j \not\perp\!\!\!\perp_\mathcal{P} X_k \mid \mathbf{X}_{\mathtt{Pre}(k,\pi)\setminus\{j\}}$$
$$\Leftrightarrow (j \to k) \in \mathtt{E}(\mathcal{G}_\pi). \qquad \text{(RU)}$$

(RU) is the method of constructing a unique DAG from $\pi$ and $\mathcal{P}$ discussed in [Raskutti and Uhler, 2018]. It is derived from a more general method in [Verma and Pearl, 1988]. The two methods will be compared in Appendix A.3. But we refer to $\mathcal{G}_\pi$ as the DAG induced from $\pi$ and the graphoid $\mathcal{P}$ using (RU) unless specified otherwise. Obviously, $\pi$ is a causal order of $\mathcal{G}_\pi$. Below is an important feature of $\mathcal{G}_\pi$.

**Definition 3.4** *(SGS-minimality) For any joint probability distribution $\mathcal{P}$, define $\mathtt{SGS}(\mathcal{P}) = \{\mathcal{G} \in \mathtt{CMC}(\mathcal{P}) : \neg\exists\mathcal{G}' \in \mathtt{CMC}(\mathcal{P})$ s.t. $\mathtt{E}(\mathcal{G}') \subset \mathtt{E}(\mathcal{G})\}$ as the set of SGS-minimal DAGs.*[2]

**Theorem 3.5** *[Verma and Pearl, 1988, Raskutti and Uhler, 2018] Given a graphoid $\mathcal{P}$ over $\mathbf{V}$, $\mathcal{G}_\pi$ induced by $\pi$ using (RU) is Markovian and SGS-minimal for every $\pi \in \Pi(\mathbf{v})$.*

The theorem above states that, for every permutation $\pi$, the induced DAG $\mathcal{G}_\pi$ is Markovian and no subgraph of $\mathcal{G}_\pi$ is Markovian. By identifying the sparsest permutation $\hat{\pi} = \mathrm{argmin}_{\pi \in \Pi(\mathbf{v})} |\mathtt{E}(\mathcal{G}_\pi)|$, $\mathcal{G}_{\hat{\pi}}$ returned by SP is guaranteed to be in $\mathtt{MEC}(\mathcal{G}^*)$ when u-frugality is satisfied. Nevertheless, SP needs to examine all $|\mathbf{v}|!$ permutations in $\Pi(\mathbf{v})$ to identify the sparsest one and hence lacks scalability. Solus et al. [2021] introduce a greedy version of SP, namely *Triangle SP* (TSP), which is proven to be correct under faithfulness.[3] Below, we provide a quick and simple sketch of this result.

TSP borrows the *Chickering algorithm* in [Chickering, 2002] to perform their *depth-first search* (DFS) procedure. For each vertex $i \in \mathbf{v}$, let $\mathtt{Pa}(i, \mathcal{G})$ be the set of *parents* in $\mathcal{G}$. A directed edge $j \to k$ is *covered* in $\mathcal{G}$ if $\mathtt{Pa}(j, \mathcal{G}) = \mathtt{Pa}(k, \mathcal{G}) \setminus \{j\}$.

**Theorem 3.6** *(Chickering sequences) [Chickering, 2002] Given a set of variables $\mathbf{V}$, for every pair of DAGs $\mathcal{G}, \mathcal{H} \in$*

---

[1]This assumption is named as *sparsest Markov representation* (SMR) in [Raskutti and Uhler, 2018].

[2]We follow Zhang [2013] to refer to this minimality condition as the one discussed in [Spirtes et al., 2000].

[3]In [Solus et al., 2021], *Greedy SP* (GSP) is an operational version of TSP which imposes a depth bound on the DFS procedure and a parameter specifying the number of runs on selecting an arbitrary initial permutation. They claimed that TSP can be correct even when faithfulness fails. We examine their claim more carefully in Section 4 and Appendix C.

DAG($\mathbf{V}$), if $\text{I}(\mathcal{H}) \subseteq \text{I}(\mathcal{G})$, *there exists a sequence of DAGs, call it a Chickering sequence* $\langle \mathcal{H} = \mathcal{G}^1, \mathcal{G}^2, ..., \mathcal{G}^k = \mathcal{G} \rangle$ *(from* $\mathcal{H}$ *to* $\mathcal{G}$*) s.t.* $\text{I}(\mathcal{G}^i) \subseteq \text{I}(\mathcal{G}^{i+1})$ *and* $\mathcal{G}^{i+1}$ *is obtained from* $\mathcal{G}^i$ *by either reversing a covered edge or deleting a directed edge for each* $1 \leq i < k$.[4]

A sequence of DAGs $\langle \mathcal{G}^1, ..., \mathcal{G}^k \rangle$ is said to be *weakly decreasing* if $|\text{E}(\mathcal{G}^i)| \geq |\text{E}(\mathcal{G}^{i+1})|$ for each $1 \leq i < k$. Obviously, every Chickering sequence is weakly decreasing. Given an arbitrary initial permutation $\pi \in \Pi(\mathbf{v})$, TSP uses DFS to search for a Chickering sequence from $\mathcal{G}_\pi$ to some SGS-minimal DAG $\mathcal{G}_\tau$ where $|\text{E}(\mathcal{G}_\pi)| > |\text{E}(\mathcal{G}_\tau)|$, and update $\mathcal{G}_\pi$ as $\mathcal{G}_\tau$ until no such $\mathcal{G}_\tau$ is found. Now we demonstrate TSP's correctness under faithfulness.

**Definition 3.7** *(U-P-minimality) For any joint probability distribution* $\mathcal{P}$*, define* $\text{Pm}(\mathcal{P}) = \{ \mathcal{G} \in \text{CMC}(\mathcal{P}) : \neg \exists \mathcal{G}' \in \text{CMC}(\mathcal{P})$ *s.t.* $\text{I}(\mathcal{G}) \subset \text{I}(\mathcal{G}') \}$ *and* $\text{uPm}(\mathcal{P}) = \{ \mathcal{G} \in \text{Pm}(\mathcal{P}) : \neg \exists \mathcal{G}' \in \text{Pm}(\mathcal{P})$ *s.t.* $\mathcal{G}' \notin \text{MEC}(\mathcal{G}) \}$ *as the sets of P-minimal DAGs and uniquely P-minimal DAGs respectively.* $(\mathcal{G}^*, \mathcal{P})$ *satisfies the u-P-minimality assumption if* $\mathcal{G}^* \in \text{uPm}(\mathcal{P})$.[5]

**Theorem 3.8** *[Zhang, 2013] For any joint probability distribution* $\mathcal{P}$*,* $\text{CFC}(\mathcal{P}) = \text{Pm}(\mathcal{P}) = \text{MEC}(\mathcal{G}^*)$ *if faithfulness holds.*

A DAG being P-minimal, as in **Definition 3.7**, states that there exists no Markovian DAG which can entail a proper superset of CI relations, and its unique variant further requires that all P-minimal DAGs belong to the same MEC as $\mathcal{G}^*$. We elaborate the importance of u-P-minimality in the next section. By **Theorem 3.6**, TSP guarantees that its output $\hat{\mathcal{G}}_\pi$ is P-minimal. When faithfulness holds, **Theorem 3.8** ensures that $\hat{\mathcal{G}}_\pi \in \text{MEC}(\mathcal{G}^*)$, and hence TSP is correct.

Notice that the identification of a Chickering sequence from $\mathcal{G}_\pi$ to a P-minimal $\mathcal{G}_\tau$ is essentially a DAG-based operation. In the next section, we introduce our permutation-based operation to converge to a P-minimal DAG, and propose a class of greedy permutation-based algorithms which employs weaker causal razors than TSP does.

In addition to TSP, Solus et al. [2021] introduced another greedy algorithm, namely *Edge SP* (ESP), which is defined by weakly decreasing traversals over the *DAG associahedron* (i.e., the *permutohedron* contracted by $\text{I}(\mathcal{P})$). These technical terms are defined in the Appendix C. ESP is shown to be assuming a weaker causal razor than TSP. In the next section, we will draw a logical discovery on how ESP is connected to our novel permutation-based operation.

---

[4]The original theorem in [Chickering, 2002] is expressed in terms of addition of directed edges. This modification helps by indicating that every Chickering sequence is a weakly decreasing sequence. In addition, one can easily observe that there does not exist any Chickering sequence from $\mathcal{H}$ to $\mathcal{G}$ if $\text{I}(\mathcal{H}) \not\subseteq \text{I}(\mathcal{G})$.

[5]P-minimality refers to the minimality condition discussed in [Pearl, 2009].

# 4 METHODS

In this section, we introduce a class of permutation-based algorithms with a generic name *Greedy Relaxations of Sparsest Permutation* (GRaSP). Three tiers of relaxation will be studied: $\text{GRaSP}_0$ is our basic algorithm, $\text{GRaSP}_1$ relaxes the search criterion of $\text{GRaSP}_0$ while $\text{GRaSP}_2$ further relaxes that of $\text{GRaSP}_1$. This hierarchy allows the identification of $\text{MEC}(\mathcal{G}^*)$ under progressively weaker causal razors. In addition, we show that $\text{GRaSP}_0$ is logically equivalent to TSP, and $\text{GRaSP}_1$ to ESP. All proofs are left in Appendix B-D. First, we introduce our characteristic permutation-based operation *tuck* and how it operates under different types of directed edges.

**Definition 4.1** *(Tuck) Consider any graphoid* $\mathcal{P}$ *over* $\mathbf{V}$*, any* $\pi \in \Pi(\mathbf{v})$*, and any* $j, k \in \mathbf{v}$ *where* $\pi[j] < \pi[k]$*. Rewrite* $\pi$ *as* $\langle \boldsymbol{\delta}_1, j, \boldsymbol{\delta}_2, k, \boldsymbol{\delta}_3 \rangle$ *where each* $\boldsymbol{\delta}_i$ *is a (possibly empty) sub-sequence of* $\pi$*.*[6] *Let* $\boldsymbol{\gamma}$ *and* $\boldsymbol{\gamma}^c$ *be the sub-sequences* $\langle i \in \boldsymbol{\delta}_2 : i \in \text{An}(k, \mathcal{G}_\pi) \rangle$ *and* $\langle i \in \boldsymbol{\delta}_2 : i \notin \text{An}(k, \mathcal{G}_\pi) \rangle$ *respectively. Define*

$$tuck(\pi, j, k) = \begin{cases} \langle \boldsymbol{\delta}_1, \boldsymbol{\gamma}, k, j, \boldsymbol{\gamma}^c, \boldsymbol{\delta}_3 \rangle & \text{if } (j \to k) \in \text{E}(\mathcal{G}_\pi) \\ \pi & \text{otherwise.} \end{cases}$$

**Definition 4.2** *Given a DAG* $\mathcal{G}$*, a directed edge* $(j \to k) \in \text{E}(\mathcal{G})$ *is said to be singular if there exists no directed path from* $j$ *to* $k$ *in* $\mathcal{G}$ *except* $j \to k$*. Define*

$$\text{E}^t(\mathcal{G}) = \begin{cases} \text{covered edges in } \text{E}(\mathcal{G}) & \text{if } t = 0 \\ \text{singular edges in } \text{E}(\mathcal{G}) & \text{if } t = 1 \\ \text{E}(\mathcal{G}) & \text{if } t = 2. \end{cases}$$

Readers can verify that $\text{E}^0(\mathcal{G}) \subseteq \text{E}^1(\mathcal{G}) \subseteq \text{E}^2(\mathcal{G})$ holds for any DAG $\mathcal{G}$. The introduction of singular edges is crucial to our logical discovery that every move ESP takes in the DAG associahedron (as defined in Appendix C) corresponds to tucking a unique singular edge. Figure 1 provides an example on how *tuck* works for each defined type of edges. As seen in the example, *tuck* is an operation that aims to change a permutation *minimally* to obtain a differently induced DAG, while a broader class of directed edges generally leads to more possible re-orderings of the vertices.

After clarifying how *tuck* works, we can define a sequence of *tuck* operations, particularly when applied to covered edges, and how a sequence of *covered tucks* (*ct*) is connected to a Chickering sequence.

**Definition 4.3** *(ct-sequence) Given a graphoid* $\mathcal{P}$ *over* $\mathbf{V}$*, for any* $\pi, \tau \in \Pi(\mathbf{v})$*,* $\tau$ *is said to be a ct-mutation of* $\pi$ *if there exist* $j, k \in \mathbf{v}$ *s.t.* $(j \to k) \in \text{E}(\mathcal{G}_\pi)$ *is covered*

---

[6]To be precise, $\boldsymbol{\delta}_1 = \langle \pi_i : 1 \leq i < \pi[j] \rangle$, $\boldsymbol{\delta}_2 = \langle \pi_i : \pi[j] < i < \pi[k] \rangle$, and $\boldsymbol{\delta}_3 = \langle \pi_i : \pi[k] < i \leq |\pi| \rangle$.

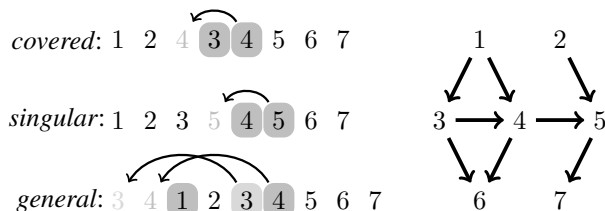

*covered*: 1 2 4 **3** **4** 5 6 7

*singular*: 1 2 3 5 **4** **5** 6 7

*general*: 3 4 **1** 2 **3** **4** 5 6 7

Figure 1: Consider $\pi = \langle 1, 2, 3, 4, 5, 6, 7 \rangle$ and its induced $\mathcal{G}_\pi$ shown on the right. Each of the three orderings on the left illustrates how a directed edge between two darkly shaded vertices is tucked to obtain a new permutation. For example, consider $1 \to 4$ which is *not* singular due to the directed path $1 \to 3 \to 4$. Performing $tuck(\pi, 1, 4)$ requires the identification of the intermediate vertices between 1 and 4 in $\pi$ which are ancestors of 4 in $\mathcal{G}_\pi$ (i.e., the lightly shaded 3). Then, while the positions of other vertices remain intact, 3 and 4 are moved to the front of 1.

and $\tau = tuck(\pi, j, k)$. Also, $\langle \pi^1, ..., \pi^m \rangle$ is said to be a ct-sequence if $\pi^{i+1}$ is a ct-mutation of $\pi^i$ for each $1 \le i < m$, and $(\mathcal{G}_{\pi^i}, \mathcal{G}_{\pi^l})$ are pairwise distinct for any $1 \le i < l \le m$.

**Lemma 4.4** *[Appendix B] Given a graphoid $\mathcal{P}$, for any $\pi \in \Pi(\mathbf{v})$ and any Chickering sequence from $\mathcal{G}_\pi$ to some $\mathcal{H} \in \text{SGS}(\mathcal{P})$ considered by TSP, there exists a ct-sequence $\langle \pi, ..., \tau \rangle$ s.t. $\mathcal{G}_\tau = \mathcal{H}$.*

Similar to the DAG-based DFS over Chickering sequences employed by TSP, the lemma above motivates our permutation-based DFS over ct-sequences as shown in **Algorithm 1**.

---

**Algorithm 1:** DFS: $dfs(\mathcal{P}, \pi, d, d_{cur}, t)$

---

**Input:** (a) $\mathcal{P}$: a graphoid over $\mathbf{V}$; (b) $\pi \in \Pi(\mathbf{v})$; (c) $d$: depth bound; (d) $d_{cur}$: recorder of the recursive call; (e) $t$: type of directed edges
**Output:** $\tau \in \Pi(\mathbf{v})$ where $score(\tau) \ge score(\pi)$

1  **foreach** $(j \to k) \in \text{E}^t(\mathcal{G}_\pi)$ **do**
2     $\tau \leftarrow tuck(\pi, j, k)$
3     **if** $score(\tau) = score(\pi)$ *and* $d_{cur} < d$ **then**
4        $\tau \leftarrow dfs(\mathcal{P}, \tau, d, d_{cur} + 1, t)$
5     **if** $score(\tau) > score(\pi)$ **then**
6        **return** $\tau$
7  **return** $\pi$

---

First, we use *negative edge count* as the scoring function in our oracle version of the algorithm such that $score(\pi) = -|\text{E}(\mathcal{G}_\pi)|$ where $\mathcal{G}_\pi$ is induced from $\pi$ and $\mathcal{P}$. $d$ bounds the search depth of DFS. We assume that $d = |\mathbf{v}|!$ for now and call the corresponding algorithm *unbounded*. We will examine some small number $d$ in light of finite samples in Section 5.2. Also, we assume that no induced DAG can be

---

**Algorithm 2:** GRaSP$_t$: $grasp(\mathcal{P}, \pi, d, t)$

---

**Input:** (a) $\mathcal{P}$: a graphoid over $\mathbf{V}$; (b) $\pi \in \Pi(\mathbf{v})$; (c) $d$: depth bound; (d) $t$: tier of GRaSP
**Output:** $\tau \in \Pi(\mathbf{v})$ where $score(\tau) \ge score(\pi)$

1  **if** $t \ne 0$ **then**
2     $\pi = grasp(\mathcal{P}, \pi, d, t - 1)$
3  $\tau \leftarrow \pi$
4  **do**
5     $\pi \leftarrow \tau$
6     $\tau \leftarrow dfs(\mathcal{P}, \pi, d, 1, t)$
7  **while** $score(\tau) > score(\pi)$
8  **return** $\tau$

---

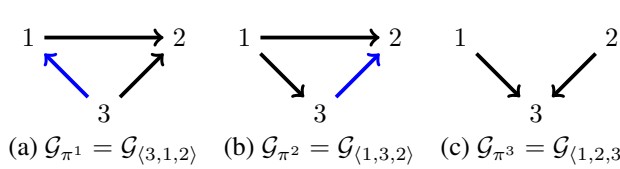

(a) $\mathcal{G}_{\pi^1} = \mathcal{G}_{\langle 3,1,2 \rangle}$    (b) $\mathcal{G}_{\pi^2} = \mathcal{G}_{\langle 1,3,2 \rangle}$    (c) $\mathcal{G}_{\pi^3} = \mathcal{G}_{\langle 1,2,3 \rangle}$

Figure 2: Example of a ct-sequence $\langle \pi^1, \pi^2, \pi^3 \rangle$ where $\text{I}(\mathcal{P}) = \{ \langle X_1, X_2 \,|\, \varnothing \rangle \}$. The blue (covered) edges indicate how a subsequent permutation is obtained by *tuck*. For example, $3 \to 1$ in (a) specifies that $\pi^2$ is obtained from $tuck(\pi^1, 3, 1)$. Also, **Algorithm 1** returns $\pi^3 = \langle 1, 2, 3 \rangle$ since the DAG in (c) is sparser than those in (a) and (b).

revisited in the DFS procedure in order to avoid any infinite loop between DAGs.

Next, $\text{E}^t(\mathcal{G}_\pi)$, as defined in **Definition 4.2**, is the crucial function distinguishing our three tiers of GRaSP in **Algorithm 2**. Consider $t = 0$ in particular. Given an arbitrary initial permutation $\pi$, **Algorithm 1** performs a greedy procedure to identify a ct-sequence from $\pi$. Figure 2 shows a simple example. Then **Algorithm 2** iterates the DFS in **Algorithm 1** until no sparser permutation can be found. Let $\hat{\tau}$ be the output of **Algorithm 2** where $\mathcal{G}_{\hat{\tau}}$ is the induced DAG accordingly. The theorem below ensures that $\mathcal{G}_{\hat{\tau}} \in \text{Pm}(\mathcal{P})$.

**Theorem 4.5** *[Appendix B] Given a graphoid $\mathcal{P}$ over $\mathbf{V}$ and any $\pi \in \Pi(\mathbf{v})$, if $\mathcal{G}_\pi \notin \text{Pm}(\mathcal{P})$, then there exists a ct-sequence $\mathfrak{T} = \langle \pi, ..., \tau \rangle$ s.t. $\mathcal{G}_\tau \in \text{Pm}(\mathcal{P})$.*

By **Theorem 4.5**, the correctness of unbounded GRaSP$_0$ under faithfulness follows immediately from **Theorem 3.8**. As shown by Forster et al. [2020], $\text{CFC}(\mathcal{P}) = \text{Fr}(\mathcal{P})$ holds under faithfulness. Since **Algorithm 2** requires that the permutation returned by a higher tier of GRaSP cannot be denser than that returned by a lower tier, the correctness of unbounded GRaSP$_1$ and unbounded GRaSP$_2$ under faithfulness immediately follows. The sample version of GRaSP can be obtained by substituting the graphoid $\mathcal{P}$ with an

*i.i.d.* observational dataset $\mathcal{D}$, and $score(\pi)$ with the BIC score of $\mathcal{G}_\pi$ from $\mathcal{D}$ (defined in Appendix E). Pointwise consistency under faithfulness directly follows from the *local consistency* of BIC.[7]

**Corollary 4.6** *Unbounded GRaSP$_0$, GRaSP$_1$, and GRaSP$_2$ are correct and pointwise consistent under faithfulness.*

Next, we want to highlight two logical discoveries with respect to the discussion of TSP and GRaSP$_0$.

**Theorem 4.7** *[Appendix B] Given a graphoid $\mathcal{P}$ and an initial permutation, the DAG returned by TSP is the same as the DAG induced by the output of unbounded GRaSP$_0$.*

The theorem above suggests that TSP and GRaSP$_0$ are *logically equivalent*. Additionally, contrary to what Solus et al. [2021] argued, faithfulness is a necessary condition for TSP.

**Theorem 4.8** *[Appendix B] Given a graphoid $\mathcal{P}$, faithfulness is necessary for the correctness of TSP.*

This theorem is entailed by a novel logical result that $\mathtt{CFC}(\mathcal{P}) = \mathtt{uPm}(\mathcal{P})$ as proven in Appendix B. Thus, the two theorems together prompt the usage of GRaSP with a higher tier. Extending $\mathtt{E}^0(\cdot)$ to $\mathtt{E}^1(\cdot)$ and $\mathtt{E}^2(\cdot)$ licenses a higher tier of GRaSP to attain a strictly sparser permutation under unfaithfulness. Examples of this sort will be studied in Section 5.1 and Appendix D.

**Corollary 4.9** *Given a graphoid $\mathcal{P}$, unbounded GRaSP$_2$ is correct under a strictly weaker causal razor than unbounded GRaSP$_1$, which is correct under a strictly weaker causal razor than unbounded GRaSP$_0$.*

Further, in Appendix C, we show the logical equivalence between unbounded GRaSP$_1$ and ESP. As a consequence, unbounded GRaSP$_2$ is a relaxation beyond the two causal razors discussed in [Solus et al., 2021]. That said, we are aware of cases where unbounded GRaSP$_2$ is incorrect under u-frugality. Such a counterexample will be studied in Section 5.1 and Appendix D.

We conclude this section by discussing how to use the DAG-inducing method in [Verma and Pearl, 1988] based on BIC scores. This facilitates our simulations done in Section 5.2. Given a semigraphoid $\mathcal{P}$ over $\mathbf{V}$, each $\pi \in \Pi(\mathbf{v})$ induces a DAG $\mathcal{G}_\pi$ satisfying the following condition:

$$X_j \in \mathbf{M} \Leftrightarrow (j \to k) \in \mathtt{E}(\mathcal{G}_\pi) \qquad \text{(VP)}$$

where $\mathbf{M}$ is a *Markov boundary* of $X_k$ relative to $\mathbf{X}_{\mathtt{Pre}(k,\pi)}$ (defined in Appendix A.3). **Lemma A.4** highlights that the

DAGs induced by (VP) and (RU) are equivalent when $\mathcal{P}$ is a graphoid. But (VP) is preferred since we can estimate the *unique* Markov boundary by the *Grow-Shrink* (GS) algorithm from [Margaritis and Thrun, 1999] using BIC scores and avoid hypothesis testing needed in (RU). We leave the discussion of the GS algorithm in Appendix E. In Section 5.2, we are going to evaluate the performance of GRaSP through (VP) and GS in light of finite samples.

# 5 SIMULATIONS

In this section, we review empirical results of unfaithful u-frugal models with respect to DAGs and algorithmic performance on Gaussian distributed data generated under a variety of situations. References to the code and instantiated models with replicability instructions are included on a GitHub site for the project[8]. Also referenced will be a running version of GRaSP in the Tetrad project (Ramsey et al. [2018]) as well as tabular data for all simulations. A scalable Python translation of GRaSP$_2$ using (VP) with a linear, Gaussian BIC score is included in the causal-learn Python package.[9]

## 5.1 U-FRUGAL FAITHFULNESS VIOLATIONS

In what follows, we consider three sets of u-frugal models that violate faithfulness. The sets of models correspond to: regular Gaussian distributions over four variables [Šimeček, 2006a], discrete distributions over four variables satisfying the intersection graphoid axiom and the Spohn condition (this includes all positive discrete distributions) [Šimeček, 2006b] (see Appendix A.2), and unfaithful DAGs (uDAGs) over five variables where a path cancellation induces a marginal independence between a pair of variables (see Appendix G)[10]. In Table 1, these sets are denoted Gaussian, Discrete, and uDAGs, respectively.

We evaluate the capabilities of GRaSP$_0$, GRaSP$_1$, and GRaSP$_2$ to recover u-frugal DAGs using an independence oracle on models from each set. We say that a GRaSP variant recovers the u-frugal model if it can do so from every permutation; if the algorithm can reach the u-frugal model from every permutation, then the correctness of the variant will be independent of the DFS implementation.

Table 1 provides a computational proof that there are GRaSP$_1$ models not found by GRaSP$_0$, and GRaSP$_2$ models not found by GRaSP$_1$. These results support the claims in **Corollary 4.9**.

---

[7]See [Haughton, 1988] and [Chickering, 2002] for the (local) consistency of BIC.

[8]https://github.com/cmu-phil/grasp.

[9]https://github.com/cmu-phil/causal-learn.

[10]The first two sets of models can be found at http://5r.matfyz.cz/skola/models.

| | GRaSP$_0$ | GRaSP$_1$ | GRaSP$_2$ | Total |
|---|---|---|---|---|
| Gaussian | 0 | 7 | 10 | 10 |
| Discrete | 0 | 79 | 84 | 84 |
| uDAGs | 0 | 19 | 49 | 61 |

Table 1: The number of u-frugal models recovered by GRaSP$_0$, GRaSP$_1$, and GRaSP$_2$ from three sets of u-frugal models that violate faithfulness. A model is considered to be recovered if it is recovered from every permutation.

## 5.2 LINEAR GAUSSIAN SIMULATIONS

We studied GRaSP's performance in the linear Gaussian case by varying simulations parameters around a configuration with 60 variables, an average degree of 6, and a sample size of 1,000 against two standard algorithms: fGES [Chickering, 2002, Ramsey et al., 2017] and PC [Spirtes et al., 2000]. In Figure 4, we vary the number of measured variables from 20 to 100 with values 20, 30, 40, 50, 60, 70, 80, 90, and 100. In Figure 3, we vary the average degree from 2 to 10 with values 2, 3, 4, 5, 6, 7, 8, 9, and 10. For Figure 5, we vary the sample size from 200 to 100,000, with values 200, 500, 1,000, 2,000, 5,000, 10,000, 20,000, 50,000, and 100,000. In all cases, we draw coefficient values uniformly from $U(-1, 1)$ and incorporate independent additive exogenous noise distributions set to $N(0, 1)$. All statistics are averaged over 20 independent runs. Finally, in Figure 6, we give the running times for our Java implementation of the algorithms. All of the algorithms except PC used BIC with a parameter penalty multiplier of 2 as a score; PC used partial correlation with a significance threshold of 0.001 as a conditional independence test. For the GRaSP variants, we allow tucks of covered edges up to depth 3, and tucks of non-covered edges at depth 1 when applicable[11]. In all cases, we follow the procedure set out in the text of running lower tiers of GRaSP before running higher tiers of GRaSP to guarantee consistent improvement of statistics.

In these figures, precision = $TP/(TP + FP)$ and recall = $TP/(TP + FN)$, where $TP$ is the number of true positives, $FP$ is the number of false positives, and $FN$ is the number of false negatives. We give precision and recall statistics for adjacencies and arrowheads separately. For adjacencies, true (false) adjacencies are pairs of vertices that are (not) adjacent in the generative graphical model, and positive (negative) adjacencies are pairs of vertices that are (not) adjacent in the estimated graphical model for each algorithm, respectively. For arrowhead statistics, a true arrowhead is a directed edge in the CPDAG[12] of the generative graphical

---

[11]In the Java implementation of the algorithm, we include parameters for uncovered depth and non-singular depth to provide the user with more control over this heuristic.

[12]A CPDAG (a.k.a. "pattern") is a graphical representation of the Markov equivalence class for a DAG. See [Spirtes et al., 2000] for details.

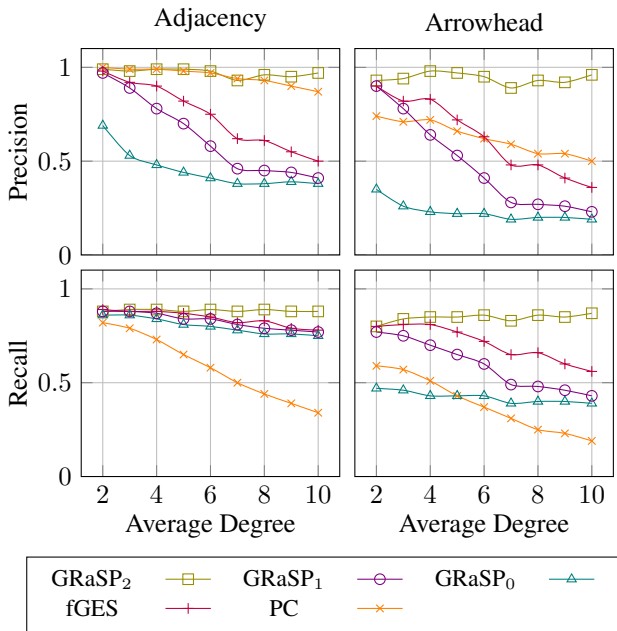

Figure 3: Average degree varied, measured variables fixed to 60, sample size fixed to 1,000.

model and a positive arrowhead is a directed edge in the CPDAG of the estimated DAG, with negative and false arrowheads indicating the absence of these directed edges in their respective CPDAGs.

Figure 3 shows that algorithmic performance is strongly dependent on the average degree. While the compared algorithms generally perform well on sparse models, their performance drops off as the density increases. The exception is GRaSP$_2$, which dominates this group of algorithms, with a strong performance for both adjacencies and arrowheads as average degree is increased.

Figure 4 shows the result of varying the number of measured variables. Notably, increasing the number of measured variables while holding the average degree constant decreases graph density. We see upward trends for some arrowhead statistics corresponding to this decrease in density. Again, GRaSP$_2$ dominates this group of algorithms, with strong precision and recall for both adjacencies and arrowheads.

All compared algorithms claim pointwise consistency, however, as shown in Figure 5, GRaSP$_2$ outputs (nearly) correct models at much smaller sample sizes; the alternative methods output incorrect models even with 100,000 samples. This might suggest that GRaSP$_2$ is better equipped to handle almost-violations of faithfulness in linear Gaussian models. As with previous figures, GRaSP$_2$ dominates this group of algorithms for precision and recall for both adjacencies and arrowheads for all sample sizes studied.

Figure 6 shows that all the algorithms on average return in under two minutes for the studied scenarios. However,

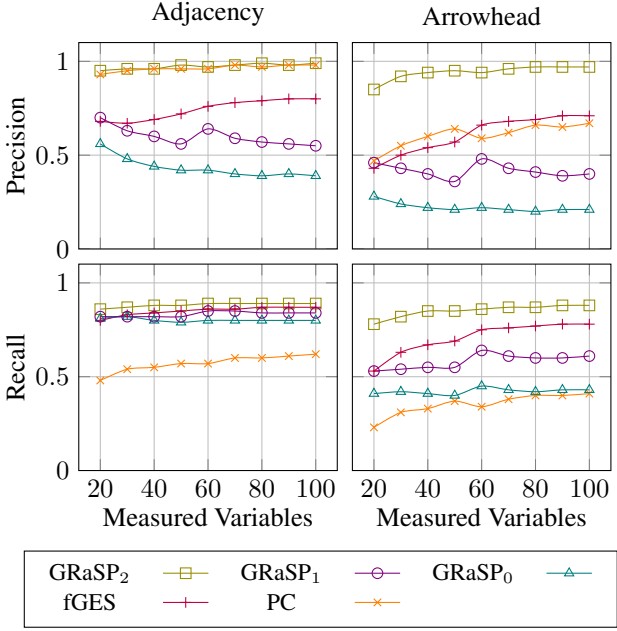

Figure 4: Measured variables varied, average degree fixed to 6, sample size fixed to 1,000.

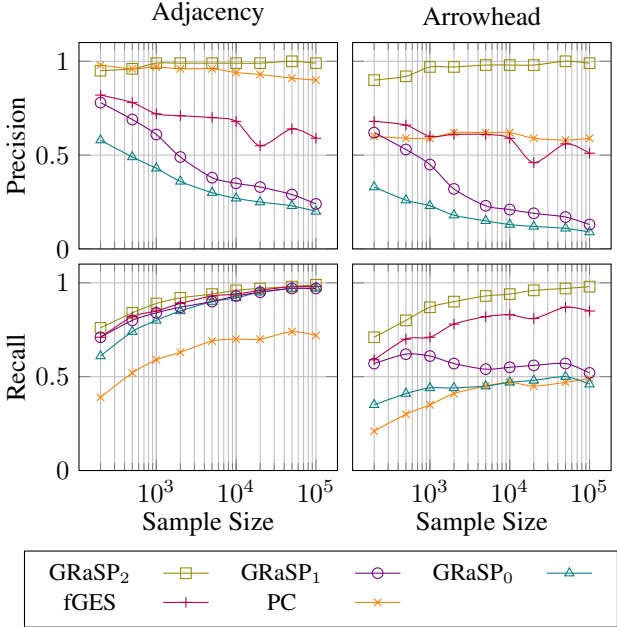

Figure 5: Sample size varied, measured variables fixed to 60, average degree fixed to 6.

given the log scale, it should be noted that the computation time for $GRaSP_2$ increases exponentially with respect to the average degree of the graph and with respect to the number of measured variables. Other algorithms see similar slow-downs, but, other than $GRaSP_1$, none of the other algorithms experience as significant of a slow-down.[13]

---

[13]All simulations in this paper were run on a MacBook Pro

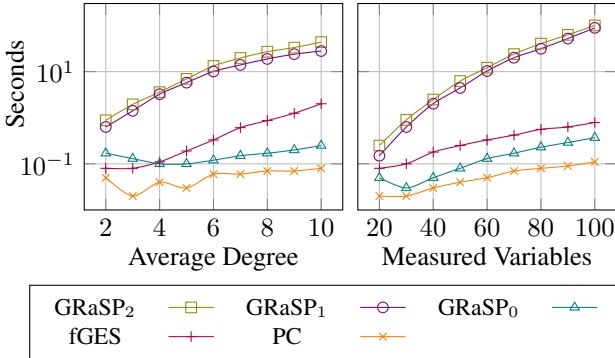

Figure 6: Measured variables fixed to 60 when not varied, average degree fixed to 6 when not varied, sample size fixed to 1,000.

In this paper, we focused on algorithms that can run on a 100 variable problem in a reasonable amount of time on a laptop. However, we would be remiss if we did not mention a recent algorithm by Lu et al. [2021] called Triplet A* that performs in terms of accuracy as well as, if not better than, $GRaSP_2$. We declined to directly compare the Triple A* algorithm in our Figures because it was unable to finish our simulations in reasonable time; for instance, the point they give in their Figure 6 for the 60-variable, average degree 5 case was already as slow as could be managed (personal communication); we took our simulations out to an average degree of 10. In lieu of this, we include in Appendix F.1 results of running $GRaSP_2$ on their published simulation data.

## 6 EMPIRICAL EXAMPLE

We give a simple empirical example, the 6-variable Airfoil example from the Irvine Machine Learning Repository (Dua and Graff [2017]. The experiment measures sound pressure elicited by an airfoil in a wind tunnel. The variables in the example are as follows: (1) Velocity of the wind in the tunnel, (2) chord length of the airfoil, (3) angle of attack of the airfoil, (4) displacement of the wind away from the airfoil, (5) frequency of the elicited sound, and (6) measured pressure of the elicited sound. (1), (2), and (3) are experimental variables and thus exogenous; (6) in the experiment is endogenous. The $GRaSP_2$, PC and GES graphs are given in Appendix F.2. The $GRaSP_2$ model (which is the same as the SP model) is uniquely frugal; background knowledge is satisfied, except possibly for (3), which looks to be not exogenous in the model; here, it helps to remember that

---

laptop computer, M1, 2020, with 16G of RAM, using the Corretto 18 Java SDK. Memory is the main resource constraint on the procedure, which is needed for caching scores. Thanks to the comment of an anonymous reviewer, a machine with 256GB of RAM may be useful for analyses significantly larger than the ones studied.

latent variables might exist. This raises the question as to whether a causally insufficient algorithm might find a model consistent with (3) being exogenous. We will explore how GRaSP$_2$ may be used to do latent variable reasoning to see whether (3) remains non-exogenous in general.

This example has a number of advantages: (a) It is an experiment so readily interpretable as a causal system; (b) because it is an experiment, partial ground truth for the system can easily be adduced; and, (c) it is small enough to run SP on the data, and since this produces a single model, we can simply compare the output of GRaSP$_2$ to the output of SP to show that GRaSP$_2$ finds the optimal BIC model.

Further empirical examples with SP (where possible), GRaSP$_2$, fGES, and PC are given on our GitHub site.

# 7   DISCUSSION

Permutation-based reasoning in designing causal search algorithms is increasingly influential in the literature, including the methods from Teyssier and Koller [2005] and Raskutti and Uhler [2018]. We propose a class of algorithms under the generic name GRaSP characterized by an efficient permutation-based operation, *tuck*. All tiers of GRaSP are shown to be correct and pointwise consistent under the assumption of faithfulness. Also, we show that the two lower tiers of GRaSP are logically equivalent to the algorithms TSP and ESP discussed in [Solus et al., 2021]. We further prove that the final tier of GRaSP makes a strictly weaker assumption than its lower-tier counterparts and demonstrate that it outperforms the lower-tier algorithms and two standard causal search algorithms, PC and fGES, in simulations.

Discussion of GRaSP can be extended in several directions. First, we have already begun to explore even higher tiers of GRaSP which relax the search criterion even further. Figure 3 suggests that GRaSP may provide tools helpful for the discussion of dense graph search. Given the hierarchy of GRaSP, higher tiers will hopefully improve the performance statistics and employ weaker assumption than the existing tiers. Ultimately, we hope to develop a tier of GRaSP that is correct under u-frugality alone.

Second, many advances have been made in the area of more or completely general modeling of data distributions, with corresponding improvements in accuracy of causal search for algorithms taking general modeling assumptions into account. It would be helpful to consider how such ideas can be incorporated into GRaSP. For example, Huang et al. [2018] show how a consistent general score can be incorporated into GES; it will be interesting to see whether GRaSP is able to show similar improvement in applicability when using such a score.

Third, we have analyzed Gaussian simulations in Section 5, but some simulation work needs to be done to show that GRaSP works well for discrete distributions (where the theory is already applicable) and also for mixed Gaussian/discrete distributions studied in [Andrews et al., 2019].

Fourth, the discussion of this paper is built upon the assumptions of causal sufficiency, that is, no latent common causes, and no selection bias. Causal search without these assumptions was pioneered by the FCI algorithm from Spirtes et al. [2000] and Zhang [2008]. To improve empirical performance of FCI, Ogarrio et al. [2016] initiated a hybrid algorithm GFCI which combines GES with FCI. To follow suit, we plan to explore an algorithm that incorporates GRaSP into GFCI (in place of GES), further improving this empirical performance.

Fifth, more direct comparisons to other algorithms need ideally to be done. As a step in this direction, we include figures on our GitHub site using the simulation parameters in [Lu et al., 2021], corresponding to their Figures 6, so there is oblique comparison to the algorithms in those figures, including GES and PC in the PCALG package [Kalisch et al., 2012], Triplet A* [Lu et al., 2021], NOTEARS [Zheng et al., 2018], the GSP implementation in the Python causaldag package, LiNGAM [Shimizu et al., 2006], and MMHC [Tsamardinos et al., 2006]. The reader is invited to explore those comparisons.

Finally, we have taken up just one real data example in this paper, but it is useful to point out in a forward-looking way that improvements in the ability to handle latent and mixed continuous/discrete variables in a scalable and accurate causal search algorithm would put one in a good position to analyze a number of otherwise difficult real data examples. Accurate preliminary results consistent with ground truth using the suggested modification of GFCI for a number of mixed datasets from the Irvine Machine Learning Repository ([Dua and Graff, 2017]), for instance, suggest that this would be a good direction to look for new practical methods (cf. [Raghu et al., 2018]).

**Author Contributions**

WL contributed theoretical results, with input from BA, while BA and JR worked on the algorithm implementations and contributed empirical results. All authors contributed to algorithmic development.

**Acknowledgements**

We thank Greg Cooper, Clark Glymour, Ignavier Ng, Peter Spirtes, Jiji Zhang, and Kun Zhang for discussion and feedback, and the anonymous reviewers for detailed and insightful comments.

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
