# OpenReview forum: "Greedy Relaxations of the Sparsest Permutation Algorithm"
_auai.org/UAI/2022/Conference — UAI 2022 Poster_

### Official Review · Reviewer_4yeM · 2022-04-13

**Q2(1) Originality/Novelty:** 3
**Q2(2) Significance/Impact:** 3
**Q2(3) Correctness/Technical Quality:** 3
**Q2(6) Clarity Of Writing:** 2
**Q6 Overall Score:** 6
**Q8 Confidence In Your Score:** 5

**Q1 Summary And Contributions:**

Authors propose alternative permutation-based causal discovery algorithms. The main contribution is the definition of a new mathematical operation to traverse the set of permutations that lead to shorter sequences than the classical method of Chickering. Authors also propose a family of algorithms by using a modification of this operation depending on what kind of edges to move around/flip.


**Q2 Assessment Of The Paper:**

More detailed information regarding each of these aspects is given below:

**Q2(4) Quality Of Experiments (Optional):**

3: Good: The experimental evaluation is adequate, and the results convincingly support the main claims.

**Q2(5) Reproducibility:**

3: Good: Key resources (e.g., proofs, code, data) are available and key details (e.g., proofs, experimental setup) are sufficiently well-described for competent researchers to confidently reproduce the main results.

**Q3 Main Strengths:**

There are some nice ideas and the authors tried to make it as self-consistent as possible with an extensive appendix. I appreciate that. Simulations show improvement over synthetic data.

**Q4 Main Weakness:**

- The paper reads Definition, Theorem, Definition with little to no text in between which leaves it to the reader's guess to fill in the gaps. I think the presentation can be greatly improved by adding some redundancy to the overall paper. For example, Theorem 2.8 has not been mentioned ever. There are also parts where the line between new contribution vs. existing work is blurred.
- Synthetic simulation results seem extensive enough although it would be good to have real-data experiments.
- A discussion on why shorter sequences would lead to better/faster algorithms is needed. It is not clear that traversing shorter sequences will be beneficial if one has to perform the same sparsification operations (see detailed comments below).


**Q5 Detailed Comments To The Authors:**

Introduction could benefit from some citations for the made claims.

It might be better to explain what the statement "While SP can be very accurate and sensibly interpreted" means.

What does "weakly decreasing sequences of adjacent transpositions" mean?

"Notably, SP, ESP, and TSP all assume the data generating distribution is a graphoid, which simplifies the computation of DAG construction considerably, since it is no longer NP-hard and can be done in polynomial time."
Could you explain how the graphoid assumption is related to solving an NP hard problem? I believe too many statements are squeezed into one sentence here and should be unpacked. What exactly was NP hard, maximum likelihood estimation?

uFrugality simply means that ALL the sparsest graphs that the pdf is Markov to, must belong to the same equivalence class of DAGs. Correct? It will help to explain verbally in more detail for clarity. Could you add an example showing why this is weaker than faithfulness?

A similar discussion would help with Definition 2.7.

it might help with clarity to use different scripts, such as i,j for \pi_i and pi[i].

RU condition does seem to require faithfulness in order for one of the DAGs to be correct. Is this true? What is the difference with Verma and Pearl? It looks like the identical construction.

Why is Theorem 2.5 presented as if it's a new result? This is proven by Pearl is it not? Proof in appendix also suggests this is not new. It probably shouldn't be called a theorem.

Thm 2.8 a. should the operators be \subset or \subseteq. if the latter, how do we know one is weaker? Could you give or point to some examples?

How does relaxing the search criterion progressively leads to weaker assumptions? Sounds like it should be the opposite. [even after finishing the whole paper, I am not fully confident about the play between assumptions and the hierarchy of algorithms. Can you elaborate on how using different kinds of arrows in the tuck operation map to having different assumptions?]

Example B.9 should be brought into the main paper, I believe. The example is enlightening but it leads to the following question:

Authors define a potentially shorter sequence for transforming one DAG to another. This is fine. But the question is is it really clear that this will provide any computational gain. The reason comes from Example B.9. In order to arrive at the conclusion that for this order in (d) we won't have an edge between 1 and 2, one still needs to check independence and weed out the edges. Chickering also does this, albeit in an extra step. So why would an abstract sequence being shorter translate to less number of operations given this? Some clarification from the authors would be greatly appreciated.

In other words, it is not clear that the following implication is true:
"We will also discuss when a ct-sequence is strictly shorter than its corre- sponding Chickering sequence in Appendix B. Hence, this suggests that the DFS of the former is more efficient than that of the latter."

Could you explain this in more detail?:
"Also, we assume that no permutation can be revisited in the DFS procedure."

**Q7 Justification For Your Score:**

The main reason for my accept score not being stronger is the presentation. It is overly compact making it hard to read. I did parse most of the claims but the content can be made more appealing to non-experts. The second reason is that I would like to see a discussion on the expected utility of shorter ct-sequences from a theoretical perspective. Why are they more advantageous? I wasn't able to locate a detailed justification for this claim.

**Q9 Complying With Reviewing Instructions:**

1: Yes.

---

### Official Review · Reviewer_iKsZ · 2022-04-13

**Q2(1) Originality/Novelty:** 3
**Q2(2) Significance/Impact:** 3
**Q2(3) Correctness/Technical Quality:** 4
**Q2(6) Clarity Of Writing:** 4
**Q6 Overall Score:** 6
**Q8 Confidence In Your Score:** 5

**Q1 Summary And Contributions:**

The authors presented a good summary of the assumptions for existing methods that aims to enable causal discovery/learning when (strong-)faithfulness is violated. The authors also presented a new search operation, tuck, which is used in a few implementation of SP-type algorithm, but not before formalized with respect to learning correctness. Using the tuck operation, the authors proposed three levels of SP algorithm that hierarchically relies on weaker assumptions and increasing reliability.

**Q2 Assessment Of The Paper:**

More detailed information regarding each of these aspects is given below:

**Q2(4) Quality Of Experiments (Optional):**

3: Good: The experimental evaluation is adequate, and the results convincingly support the main claims.

**Q2(5) Reproducibility:**

4: Excellent: Key resources (e.g., proofs, code, data) are available and key details (e.g., proof sketches, experimental setup) are comprehensively described for competent researchers to confidently and easily reproduce the main results.

**Q3 Main Strengths:**

This paper gives a good summary of the various (often not unified) weaker-than-faithfulness notions of "requirement of graph/distribution" for successful causal learning. The writing is very high quality and detail-oriented, very readable and would be a good reference point for future research. All theoretical claims are backed with reference and clear proofs. The simulation results nicely demonstrated the performance of the proposed method at different levels of generalization.

**Q4 Main Weakness:**

For me the main weakness is on lack of novelty. Though the generalization is new and the "tuck" operation is not seen in previous discussions on SP-type algorithms, they are more like implementation tricks than new method. Specifically, for the three levels of generalization, the 0th level is simply the same as TSP, whereas 1st and 2nd levels are just "search more edges" versions of it. There are no clean theoretical results on what the learned object from 1st and 2nd level relaxation. And specifically for the 2nd level relaxation, I my experience (I have coded up a very similar program before) this would be computational intensive for any graph larger than 200 nodes and edge-degree >5, so it's not as good as seems either. As for simulation, I do think it is important to compare with triplet a* no matter what the results may look

This being said, I do think that this result (with added simulation) maybe interesting to the causal structure learning community and open to change my score

**Q5 Detailed Comments To The Authors:**

writing:
- Page 2 “All known causal search algorithms that are correct assume the Markov assumption”: should probably define what you mean by causal search and which among them are “classical” methods
- Alg 1. Missing argument for t

method:
- I would like to see the 0/1/2 algorithms linked with their own theoretical results, e.g., what is the learned object of generalization level t, and how are the guarantee differs when making 0/1/2 assumptions. My guess is that the result will not be as elegant as uPM etc (basically, at level t you can learn what you can learn at level t?) If such results are not shown, then these "relaxations" are really just implementation tricks to nudge more precision out of TSP.
- simulation. Yes I do think comparing with triplet A* is important, due to the fact that similar ideas (shorter traverse length) is used in that paper.

**Q7 Justification For Your Score:**

Good paper, great writing. My only concern is lack of novelty and I'm happy to change my score if other reviewers think otherwise.

**Q9 Complying With Reviewing Instructions:**

1: Yes.

---

### Official Review · Reviewer_tS9z · 2022-04-13

**Q2(1) Originality/Novelty:** 3
**Q2(2) Significance/Impact:** 3
**Q2(3) Correctness/Technical Quality:** 3
**Q2(6) Clarity Of Writing:** 3
**Q6 Overall Score:** 7
**Q8 Confidence In Your Score:** 3

**Q1 Summary And Contributions:**

The paper presents a greedy causal search algorithm in the space of permutations. The main claim is that the proposed greedy algorithm is consistent (i.e. finds the true graph) under an assumption strictly weaker than the causal faithfulness condition used in other approaches (GES,..) and scales to larger graphs (up to 100 variables) than its competitors having the same consistency property. It compares favourably to GES and PC algorithms on some benchmarks.



**Q10 Ethical Concerns (Optional):**

No.

**Q2 Assessment Of The Paper:**

More detailed information regarding each of these aspects is given below:

**Q2(4) Quality Of Experiments (Optional):**

3: Good: The experimental evaluation is adequate, and the results convincingly support the main claims.

**Q2(5) Reproducibility:**

3: Good: Key resources (e.g., proofs, code, data) are available and key details (e.g., proofs, experimental setup) are sufficiently well-described for competent researchers to confidently reproduce the main results.

**Q3 Main Strengths:**

- consistent algorithm with weaker assumptions than the causal faithfulness condition (GRaSP1 or GRaSP2)
- new swap operator in the context of permutations
- GRaSP0 can perform fewer iterations than GES
- on very small and unfaithful graphoids, GRaSP2 finds the true graph whatever the initial permutation
- good results compared to GES and PC on linear Gaussian models


**Q4 Main Weakness:**

- no specific justification for the swap operator
- consistency relies on an unbounded version with an exponential depth-first search method (O(n!))
- lots of definitions and not so many examples
- algorithm complexity not analyzed
- empirical computation time exponential in the number of variables or edges
- no experimental results with different depth bound values
- no experimental results on very large (more than 100 variables) and/or (mixed) discrete distributions with other local search algorithms (Teyssier&Koller2005, see also References)


**Q5 Detailed Comments To The Authors:**

The paper presents a greedy causal search algorithm in the space of permutations (variable orderings). It uses a particular non-adjacent swap operator applied on permutations that relies on an implied DAG. The score of a DAG aims at maximizing the number of conditional independence found in the true distribution (minimal edge count). The main claim is that the proposed greedy algorithm is consistent (i.e. finds the true graph) under an assumption strictly weaker than the causal faithfulness condition used in other approaches (GES,..) and scales to larger graphs (up to 100 variables) than its competitors having the same consistency property. A minor claim is that it can be more efficient than GES, i.e. find (local) optimal DAGs in fewer iterations. It is compared to classical GES (which is encompassed by the approach) and PC algorithms and showed (much) better reconstruction quality on linear Gaussian models and also on small (4-5 variables) graphs without the faithfulness property. Their approach was especially better on dense graphs. Computation time was less than 30 seconds on the largest 100-variable graphs.

I found the paper quite dense to read. There are not many examples. The background theory is quite large and may be compacted even more if not used later in Section 3 (but only in Appendix proofs). The theoretical results seem important and supported by reasonable good experimental results. I would have been more convinced if larger graphs and (mixed) discrete distributions would have been tried and compared with more recent structure learning algorithms.

The complexity of finding an optimal DAG from a given permutation should be exponential in the number of variables or an in-degree bound is used to limit this task. This complexity issue is not mentioned in the paper and I wonder how it is done in the experiments with 100 variables. In particular,  the non-adjacent swap operator between j and k relies on the identification of all ancestors of k and belonging to the interval between j and k in the order of the permutation.  The fact that the computation time of GRaSP2 increases exponentially in the experiments should be better analyzed in terms of algorithm complexity.

Moreover, the motivation for their non-adjacent swap operator is not explained. It seems that the new operator preserves the local optimal scores for all the variables except for j and gamma^c.

Questions
--------------

“Notably, SP, ESP, and TSP all assume the data generating distribution is a graphoid, which simplifies the computation of DAG construction considerably since it is no longer NP-hard and can be done in polynomial time.” Which reference? What does it mean in the case of a finite sample?

How do you implement the sparsest permutation score in Alg.1 and 2? (caching which scores?)

What is the impact on time and solution quality of different depth bound values in the look-ahead DFS procedure for the linear Gaussian experiments?

For large instances, what is the impact of the initial permutation? Do restarts improve solution quality?

Do the linear Gaussian models contain cycles?

Theorem 3.11 says that unbounded GRaSP0 is logically equivalent to GES under faithfulness conditions but this is not confirmed by the experiments in 4.2, GES being better than GRaSP0? Why?

References on other operators in the space of DAGs or permutations and scaling to larger problems
------------------------------------------------------------------------------------------------------------------------------------

J Vandel, B Mangin, and S de Givry. New Local Move Operators for Bayesian Network Structure Learning. In Proc. of PGM-12, Granada, Spain, 2012.

Colin Lee and Peter van Beek. Metaheuristics for Score-and-Search Bayesian Network Structure Learning. Proceedings of the 30th Canadian Conference on Artificial Intelligence, Edmonton, Alberta, May, 2017

Scanagatta, M., Corani, G. &amp; Zaffalon, M.. (2017). Improved Local Search in Bayesian Networks Structure Learning. Proceedings of The 3rd International Workshop on Advanced Methodologies for Bayesian Networks, in Proceedings of Machine Learning Research 73:45-56.

Minor remarks
-------------------
page 2. Graph G* is not clearly defined. Does it always exist?

page 3.  “the sparsest permutation \hat\pi = min” --> argmin
 “and hence GSP is correct” --> TSP

page 6. “Notice that if GRaSP t can reach the u-frugal unfaithful model from every initial permutation, then the correctness of the algorithm will be independent of the DFS implementation”. I suspect the order in which DFS proceeds may have also an impact on the results, because it applies a move as soon the score is improved and does not wait to select the best move in the neighborhood.

Page 7. Fig 2. Why precision increases when the number of variables increases for GRaSP2?
Fig 3. Why decreasing precision for fGES, GRaSP0 and GRaSP1 when the sample size increases?

Page 8. Footnote 12. No need for a supercomputer to get 256GB of memory available. Most (linux) PC servers should have such a configuration.


**Q7 Justification For Your Score:**

The theoretical results seem important and supported by reasonable good experimental results.

**Q9 Complying With Reviewing Instructions:**

1: Yes.

---

### Official Review · Reviewer_Z3sU · 2022-04-17

**Q2(1) Originality/Novelty:** 3
**Q2(2) Significance/Impact:** 2
**Q2(3) Correctness/Technical Quality:** 3
**Q2(6) Clarity Of Writing:** 3
**Q6 Overall Score:** 6
**Q8 Confidence In Your Score:** 3

**Q1 Summary And Contributions:**

This work consider order based causal search algorithms, where the aim is to return a causal DAG given an observational distribution. It introduces the GRaSP algorithm which uses an efficient operation to traverse the search space and of which one form relaxes assumptions from prior work (SP based versions). In experiments, the final tier is shown to outperform the other versions and two well-known causal methods.


**Q2 Assessment Of The Paper:**

More detailed information regarding each of these aspects is given below:

**Q2(4) Quality Of Experiments (Optional):**

3: Good: The experimental evaluation is adequate, and the results convincingly support the main claims.

**Q2(5) Reproducibility:**

4: Excellent: Key resources (e.g., proofs, code, data) are available and key details (e.g., proof sketches, experimental setup) are comprehensively described for competent researchers to confidently and easily reproduce the main results.

**Q3 Main Strengths:**

A novel method is introduced, iterating on prior work on causal discovery trough permutation, and where one tier improves over prior work, which is also shown in experiments along various axes of approach. Reproducibility is excellent with the authors releasing the source code upon publication. Writing style overall is very good, and technical quality is excellent (although most proofs are left to the appendix which I haven't completely checked).


**Q4 Main Weakness:**

The guarantees for the tier 2 method without faithfulness are not entirely clear (see Q5), and it could perhaps be seen as technically a improvement over prior methods. No (semi-)real-world experiment is included. Readability could be improved by providing intuition around certain key concepts.


**Q5 Detailed Comments To The Authors:**

Comments following paper order:
- Sec 1 "This is not completely unreasonable […] be relied upon. "Are causal discovery methods in general meant here? Could include citation here , and also here: "However, the performance of these [..] is densely connected."
- Sec 1 "More specifically [..] or no cycles" This has been relaxed in many instances, "customarily" seems strong here.
- Sec 1-2 The notion of "causal search" algorithms in sec 1 seems to refer to all causal discovery methods in general, or at least those constraint-based, while in Sec 2 this seems to mean only the permutation based ones ("causal search algorithms, which rely on assumptions strictly weaker than faithfulness")
- Sec 2 "All causal search algorithms that are correct.." -> remove "that are correct"
- end of p3 "hence GSP is correct." GSP only defined in footnote
- Sec 3 The paragraph below Thm 3.7 helped in providing intuition to the prior part of this section, mainly the theorem and the algorithms. Having this intuition in the paper around these concepts could improve readability. Intuition for Thm 3.11 is missing, in general these three Thms are posited without surrounding text.
- Sec 3 Def 3.6 what is the intuition behind GraSP_1 using only singular edges?
- Sec 3 Relatedly, is there a soundness guarantee behind (unbounded) GraSP_1 and GraSP_2 without faithfulness?
- Sec 4.1 PC would here recover 0 models too, and not a single graph but a MEC?
- Sec 4.1 Is in these experiments "running lower tiers of GRaSP before running higher tiers of GRaSP to guarantee consistent improvement of statistics." applied? And if not, why so?
- Sec 4.2 Readability of graphs can be improved by color coding each line
- Sec 4.2 The improvement in performance of GRaSP_2 over t=0 and t=1 is in some instances dramatic, could you add a comment to why the difference is quite so large?
- Sec 4.2 GRaSP_2 is shown to outperform other methods in the simulation settings. How would a similar method perform without the tiered run of GraSP_t t=0, t=1, t=2?
- Sec 5. "d demonstrate that it outperforms the lower-tier algorithms and two other state-of-the-art causal search algorithm" Given the last paragraph of Sec. 4 about the work Lu. 2021, need to add a qualifier that this holds for efficient computation.
- App F. Spelling: "properties, is symmetry assumed."

**Q7 Justification For Your Score:**

The writing in this work is of high quality, with minor comments towards presentation style. The GRaSP_2 variant of the method is a relaxation of the search space used in existing ordering methods as is convincingly seen in experiments, while the other variants are prior work. To a lesser extent soundness of GRaSP_2 is shown in the theory section under faithfulness only.


**Q9 Complying With Reviewing Instructions:**

1: Yes.

---

### Decision · Program_Chairs · 2022-05-15

**Decision:**

Accept (Poster)

**Comment:**

Meta Review: The reviewers have reached a consensus on accepting the paper. The paper makes non-trivial advances over prior work with reasonable experimental evaluation. The writing is clear overall but could be improved as suggested by the reviewers. Please revise the paper to address the issues raised by the reviewers in the camera-ready version -  the proposed revision plan in the author's responses look adequate.